# The Dually Localized EF-Hand Domain-Containing Protein TgEFP1 Regulates the Lytic Cycle of *Toxoplasma gondii*

**DOI:** 10.3390/cells11101709

**Published:** 2022-05-21

**Authors:** Noopur Dave, Kaice LaFavers, Gustavo Arrizabalaga

**Affiliations:** Department of Pharmacology and Toxicology, Indiana University School of Medicine, Indianapolis, IN 46202, USA; nkdave@iu.edu (N.D.); klafaver@iupui.edu (K.L.)

**Keywords:** *Toxoplasma*, EF-hand, calcium, egress, vacuole

## Abstract

The propagation of the obligate intracellular parasite *Toxoplasma gondii* is tightly regulated by calcium signaling. However, the mechanisms by which calcium homeostasis and fluxes are regulated in this human pathogen are not fully understood. To identify Toxoplasma’s calcium homeostasis network, we have characterized a novel EF-hand domain-containing protein, which we have named TgEFP1. We have determined that TgEFP1 localizes to a previously described compartment known as the plant-like vacuole or the endosomal-like compartment (PLV/ELC), which harbors several proteins related to ionic regulation. Interestingly, partial permeabilization techniques showed that TgEFP1 is also secreted into the parasitophorous vacuole (PV), within which the parasite divides. Ultrastructure expansion microscopy confirmed the unusual dual localization of TgEFP1 at the PLV/ELC and the PV. Furthermore, we determined that the localization of TgEFP1 to the PV, but not to the PLV/ELC, is affected by disruption of Golgi-dependent transport with Brefeldin A. Knockout of TgEFP1 results in faster propagation in tissue culture, hypersensitivity to calcium ionophore-induced egress, and premature natural egress. Thus, our work has revealed an interplay between the PV and the PLV/ELC and a role for TgEFP1 in the regulation of calcium-dependent events.

## 1. Introduction

Calcium is a ubiquitous second messenger that regulates essential cell functions, including gene expression, protein secretion, metabolism, and apoptosis in various mammalian cell types [1]. Due to the crucial role that calcium plays in these different cellular functions, calcium levels are highly regulated, and localized calcium fluxes have been shown to regulate organelle-specific functions, including oxidative metabolism at the mitochondrial matrix and gene expression in the nucleus [1,2]. Several intracellular homeostatic mechanisms are used to regulate cytoplasmic calcium levels under resting and non-resting conditions, including activation of calcium-sensing and conducting proteins, which leads to the release or uptake of calcium by major calcium reservoirs within the cell or from the extracellular milieu [1].

Not surprisingly, calcium is also an important second messenger that regulates functions essential to the growth cycle of many eukaryotic human pathogens, including the protozoan parasite *Toxoplasma gondii* [3]. *T. gondii* infects a third of the human population and can cause severe disease in the immunocompromised and those infected congenitally [4,5]. A significant portion of this parasite’s pathogenesis is due to the lytic nature of its propagation cycle. As an obligate intracellular parasite, *T. gondii* needs to be inside a cell to divide, and it propagates by repeating cycles of host cell attachment, active invasion, and egress. Previous studies have shown that calcium plays a key role in regulating key steps of the *T. gondii* lytic cycle. Specifically, parasite-specific proteins needed for attachment and invasion are secreted in a calcium-dependent manner [3]. Furthermore, there is a temporal increase in calcium levels in the host cell, the parasitophorous vacuole (PV) within which the parasites reside and divide, and the parasite cytosol, which regulates the timing of egress [6].

Extensive work has been conducted to understand calcium signaling in the parasite [6,7]. However, the mechanisms responsible for regulating calcium homeostasis and sensing are not thoroughly understood. The biggest calcium reservoir in mammalian cells is the endoplasmic or sarcoplasmic reticulum (ER/SR) [2]. The uptake of calcium from the ER/SR is regulated by the sarcoendoplasmic reticulum Ca^2+^-ATPase (SERCA) [1,2]. The release of calcium from the ER/SR is regulated by 1,4,5-triphosphate receptors (IP3R) and ryanodine receptors (RyR). Similarly, the ER of *T. gondii* has been implicated in calcium storage and calcium release. Nonetheless, while the parasite is sensitive to IP3, an IP3 receptor has not been identified [6,8].

Apart from the ER, *T. gondii* also possesses an intracellular acidocalsisome-like compartment that is predicted to be a calcium store within the parasite [9]. Previous work identified and characterized two homologous proteins in *T. gondii* that localize to the acidocalcisome in other model organisms, including a vacuolar-type H^+^ pyrophosphastase (TgVP1) and a Ca^2+^-ATPase (TgA1) [9,10,11]. Both of these proteins have been shown to play key roles in parasite calcium homeostasis [11]. Other intracellular calcium stores in eukaryotic cells include the mitochondria, nucleus, and secretory granules in excitatory cells [2], but to date, no evidence exists for these organelles playing a role in calcium fluxes in *T. gondii*.

Extensive work from our laboratory, and others, on a unique compartment known as the plant-like vacuole or the endosome-like compartment (PLV/ELC) determined that this organellar network harbors several ion binding and/or conducting proteins, including the sodium-proton exchanger 3 (TgNHE3), aquaporin (TgAQP1) and a vacuolar-pyrophosphatase (TgVP1) [10,12]. Furthermore, a calcium-proton exchanger has been predicted to localize to the PLV/ELC, suggesting that this compartment may play a role as a calcium store in the parasite [13]. However, little is known about the function of the PLV/ELC in parasite calcium homeostasis.

Interestingly, work from our laboratory, and others, has shown that the parasitophorous vacuole (PV), within which the parasite resides, may also play a key role in calcium regulation [3,14]. Through a forward genetic selection for parasites resistant to the effects of calcium ionophores, we identified GRA41, a protein secreted into the PV during parasite intracellular growth. Parasites lacking GRA41 exhibit abnormal division, calcium dysregulation, and premature egress from the host [14]. We have now identified a novel EF-hand domain-containing protein, TgEFP1, which is present in the PV during intracellular growth. Similar to GRA41, disruption of this protein affects the timing of egress and the rate of parasite propagation. Interestingly, we show that TgEFP1 is predominantly present within the PLV/ELC, suggesting an interplay between this compartment and the PV. Importantly, this work suggests that the PV and PLV/ELC collaborate in the control of the parasite propagation cycle through the regulation of ionic homeostasis.

## 2. Materials and Methods

### 2.1. Host Cell and Parasite Maintenance

All parasite strains were continuously passed through Human Foreskin Fibroblast Cells (HFF-1; ATCC Catalog # SCRC-1041^™^). Cultures were maintained in Dulbecco’s Modified Eagle’s Medium (Thermo Fischer Scientific Catalog # 11885084) supplemented with 10% heat-inactivated Fetal Bovine Serum (FBS; Corning^®^ Catalog # 35-015-CV), 100 mg streptomycin/100 U penicillin per mL, and 2 mM L-glutamine. Cultures were grown in a humidified incubator at 37 °C and 5% CO_2_. The parental strain used is the RH strain lacking genes encoding hypoxanthine-xanthine-guanine phosphoribosyltransferase (*HXGPRT*) and Ku80 (RH∆Ku80∆*hxgprt*) [15,16]. Parasite lines under pyrimethamine selection were cultured in DMEM supplemented with dialyzed FBS (Corning^®^ Catalog # 35-071-CV. Parasites were selected with pyrimethamine (1 mM), mycophenolic acid (50 mg/mL) and/or xanthine (50 mg/mL). Stock pyrimethamine and mycophenolic acid were prepared in ethanol, while xanthine was prepared in water. All drugs were purchased from Sigma-Aldrich (St. Louis, MO, USA).

### 2.2. Generation of Transgenic Parasite Lines

To generate the TgEFP1-HA parasite strain, a double homologous recombination strategy was used to add a triple hemagglutinin (HA) epitope tag to the endogenous gene. A fragment of homology upstream of the TgGT1_255660 stop codon was cloned into the PacI site of the pLIC-3xHA-DHFR vector by ligase-independent cloning. All primers used in this study are listed in Appendix A. The resulting vector was linearized with SacI restriction enzyme and transfected into RHΔKu80 parasites. Transfected parasites were selected with pyrimethamine, and clones were established by serial dilution.

To generate the TgEFP1 knockout parasites, the TgEFP1-HA expressing strain was transfected with plasmids expressing Cas9-GFP and either of two guide RNAs (sgRNA 1 and 2). These vectors were generated by mutating the sgRNA site in the pSag1-Cas9-U6-sgUPRT-HXG plasmid [16] using the Q5 Site-Directed Mutagenesis Kit (NEB). The two guide RNAs were designed to target TgEFP1 at either of the predicted EF-hand domains using the online E-CRISP tool (http://www.e-crisp.org/E-CRISP/ accessed on: 6 November 2019). Parasites were co-transfected with either of the sgRNA expressing vectors and a donor template consisting of the gene encoding HXGPRT flanked by fragments of homology to regions upstream and downstream of each sgRNA. Stable lines were established through selection with MPA and xanthine. Serial dilution was used to establish clones, and one from each of the two populations was chosen for this work: TgEFP1-KO clone 1 from those transfected with sgRNA1 and TgEFP1-KO clone 2 from the sgRNA 2 transfection. Disruption of TgEFP1 in both clones was confirmed by western blot and immunofluorescence assays.

TgEFP1-KO Clone A was generated in the RHΔKu80D*hxgprt* strain using the same CRISPR/Cas9 plasmid and donor template as for TgEFP-KO clone 1. Insertion of the template into the *TgEFP1* locus was confirmed by PCR. To generate a plasmid for complementation, we amplified the *TgEFP1* coding sequence (CDS) from genomic DNA extracted from the parental strain and a region of 1100 base pairs bases upstream of the TgEFP1 start codon, which would contain the promoter and 5′UTR of the gene. Both amplicons were cloned into the pLIC-3xHA-DHFR vector using In-Fusion HD Cloning to obtain the plasmid 5′UTR-TgEFP1-3xHA-DHFR-HXGPRT. In-Fusion^®^ Cloning was used on this plasmid to generate the signal peptide deletion mutant (SP Del), in which residues 2–27 of TgEFP1 were deleted. Q5 mutagenesis was used to generate the D97A and D129A mutant versions of TgEFP1. The TgEFP1 WT and mutant complementation plasmids were used as templates to amplify cassettes that include efp1-EFP1-HA and the HXGPRT selection maker, which were transfected into the TgEFP1-KO Clone A strain. To direct integration of the complementation cassettes into the 5’UTR of the disrupted KU80 gene in the knockout strain, we generated pSag1-Cas9-U6-sgKU80.5’UTR-HX, which expresses Cas9 and a sgRNA targeting Ku80. TgEFP1-KO Clone A parasites were transfected with both the complementation cassettes and the pSag1-Cas9-U6-sgKU80.5’UTR-HX plasmid. Parasites were selected with pyrimethamine, MPA, and xanthine. Clones were established through serial dilution. IFA was used to confirm the expression of TgEFP1 in the complemented strains.

### 2.3. Immunofluorescence Assays and Western Blot

Immunofluorescence assays (IFAs) and western blots were performed as previously described [17,18,19]. For the western blots, extracellular parasite protein extract was made from parasites allowed to naturally undergo egress. Intracellular parasite extract was from HFFs infected for 30 h. For this study, the primary antibodies used were monoclonal anti-HA rabbit and/or anti-SAG1 mouse at 1:5000, followed by goat anti-rabbit and/or goat anti-mouse conjugated to horseradish peroxidase (HRP) at 1:10,000 in non-fat dry milk in TBST (Tris buffer saline solution, 0.1% Tween 20), as previously described [17].

For immunofluorescence assays, extracellular parasites were attached to poly-lysine treated glass coverslips as previously described [19]. For immunofluorescence assays of intracellular parasites, host cells were grown on glass coverslips and subsequently infected with parasites at an MOI of 2. Samples were stained with antibodies against organelle-specific primary antibodies (1:1000), including mouse anti-TgATREX (apicoplast), mouse anti-TgSERCA (ER), mouse anti-F1β-ATPase (mitochondrion), mouse anti-TgROP6 (rhoptries), mouse anti-TgCPL (VAC), guinea pig anti-TgNHE3 (PLV), rat anti-TgSORTLR (Golgi), and mouse anti-acetylated tubulin. In addition, to detect HA-tagged proteins, we used rabbit anti-HA antibodies (1:1000). Secondary antibodies used were Alexa-Fluor 594 anti-Mouse (1:2000), Alexa-Fluor 647 anti-mouse (1:2000), Alexa-Fluor 568 anti-Guinea pig (1:2000), Alexa-Fluor 488 anti-Rabbit, Alexa-Fluor 568 anti-guinea pig, and Alexa Fluor 647 anti-rat (1:2000).

### 2.4. Brefeldin A (BFA) Experiments

Host cells were grown on 15 mm glass coverslips in 24-well plates and subsequently infected with freshly lysed parasites at an MOI of 2. After 24 h, samples were treated with 0, 1, 2.5, or 5 mM BFA in HBSS for either 15 min, 30 min, or 1 h. Samples were fixed at appropriate time points in 4% paraformaldehyde/1X PBS pH 7.4 for 15 min at room temperature. IFA was performed as described above.

### 2.5. Ultrastructure Expansion Microscopy

Host cells were seeded on 15 mm coverslips in a 24-well plate, infected with parasites for 18 h, and fixed. After fixation, ultrastructure expansion microscopy protocol was used to expand and image samples as outlined in a previous study [20]. Level of expansion was determined by measuring the diameter of the gel after expansion and comparing it to the diameter of the coverslip. Samples were stained with the following primary antibodies: rabbit anti-HA (1:500), mouse anti-TgGRA5 (1:250), and/or mouse anti-acetylated tubulin. Subsequently, samples were stained with the following secondary antibodies: goat anti-Mouse Alexa-Fluor 594, goat anti-mouse Alexa-Fluor 647, and/or goat anti-rabbit Alexa-Fluor 488. All samples were also co-stained with Alexa-Fluor 405 NHS-ester to stain for protein density.

### 2.6. Phenotypic Assays

All phenotypic assays were performed as previously described [17,19,21]. In brief, for plaque assays, HFFs grown in 12-well plates were infected with 500 freshly lysed parasites and allowed to grow for 5–6 days, after which samples were fixed with methanol and stained with crystal violet. Imaging of samples was undertaken using the Protein Simple FluorChem M system imager, and the plaque area clearance was quantified using ImageJ and the ColonyArea plugin [22]. For the egress assays, infected HFFs were treated with 0, 0.1, 0.5, or 1 μM of A23187 prepared in HBSS buffer (Gibco) for 2 min at 37 °C. Samples were then fixed with methanol and stained with Hema3 Manual Staining System (Fisher Scientific, Hampton, NH, USA). The number of intact and lysed vacuoles was recorded across all cell lines using a light microscope. For doubling assays, approximately 2 × 10^4^ freshly lysed parasites were allowed to invade HFFs for 2 h, after which cultures were washed 5 times with warm media to eliminate parasites remaining outside cells. Cultures were then grown for 24 and 30 h before fixation with methanol and staining with Hema3 Manual Staining System (Fisher Scientific). For each sample, 50 vacuoles were randomly selected, and the number of parasites per vacuole was recorded. All phenotypic assays were conducted in biological and experimental triplicates and statistically analyzed student *t*-tests assuming equal variance.

### 2.7. Statistical Analysis

All statistical analysis was performed with Microsoft Excel. All data were analyzed using Student’s *t*-test to evaluate statistical significance.

## 3. Results

### 3.1. TgEFP1-HA Is Localized to the PLV/ELC and the Parasitophorous Vacuole

To identify proteins involved in calcium regulation that might be secreted, we searched the *Toxoplasma gondii* database (ToxoDB) for proteins with both EF-hands and signal peptides. In this manner we identified TgGT1_255660, TgGT1__293760, and TgGT1_227800. User comments in ToxoDB indicate that TgGT1_293760 and TgGT1_227800 localized to the Golgi and the IMC sutures, respectively. Accordingly, we focused our investigation on the uncharacterized TgGT1_255660, a 149 amino acid protein that we refer to as TgEFP1. The online tools SignalP-5.0 and Scan Prosite predict an N-terminal signal peptide between amino acids 1 and 27 and two canonical C-terminal EF-hand domains between the amino acids 84 to 119 and 119 to 149 (Figure 1A) [23,24,25]. To identify the localization and expression pattern of TgEFP1, we introduced a C-terminal triple hemagglutinin (3xHA) epitope tag along with the selectable DHFR marker at the endogenous locus using double homologous recombination (Figure 1A). Western blot analysis probing with anti-HA primary antibody showed a specific band in the TgEFP1-HA lysate at the predicted 19 kilodaltons (kDa) size. Western blot analysis also showed that TgEFP1-HA was present in both intracellular and extracellular parasites at similar levels (Figure 1B). Immunofluorescence assay (IFA) probing with anti-HA primary antibody showed that TgEFP1-HA localized to an internal compartment within the parasite and around the parasite cell body (Figure 1C). IFA analysis of TgEFP1-HA parasites probing with anti-HA along with an array of organelle-specific antibodies showed that TgEFP1-HA only co-localized with the plant-like vacuole or endosomal-like compartment (PLV/ELC) marker TgNHE3 (Figure 1D). In contrast, TgEFP1-HA did not co-localize with other organelle markers, including TgROP6 at the rhoptries, TgCPL at the lysosome-like vacuolar compartment (VAC), TgF1β-ATPase at the mitochondrion, TgATREX at the apicoplast, TgVP1 at the acidocalcisome, and TgSERCA at the endoplasmic reticulum (Figure 1D, Appendix A). Altogether, TgEFP1-HA is expressed in both intracellular and extracellular parasites and localizes to a discrete area of the parasite where it co-localizes with TgNHE3.

Previous studies have shown that in extracellular parasites, the lysosome-like compartment named the VAC and other endosomal compartments, including the PLV/ELC, condense into one punctate region at the trans-Golgi network of the parasite [26]. The VAC then exhibits dynamic fragmentation in intracellular parasites, where it separates from the trans-Golgi network and fragments into smaller vesicles [26]. To identify if the compartment containing TgEFP1 also exhibits the same dynamic pattern, we performed IFA of intracellular and extracellular parasites probing for the trans-Golgi network marker TgSortilin-like receptor (TgSOTRTL), the known PLV/ELC marker TgNHE3, and TgEFP1-HA (Figure 2A). In intracellular parasites, the signals from all three markers appear separate from each other, where TgEFP1-HA is abutting to TgSORTLR and co-localizes with TgNHE3 (Figure 2A, top row). However, in extracellular parasites, TgEFP1-HA co-localizes with TgSORTLR, TgNHE3, and TgEFP1-HA signals (Figure 2A, bottom row). Thus, similar to other parasite endosomal compartments, the compartment harboring TgEFP1 and TgNHE3 exhibits dynamic events that include coalescence in extracellular parasites.

Closer observation of immunofluorescence images showed that the TgEFP1-HA localization pattern differed between different vacuoles. This could be due to dynamic localization during parasite division. *T. gondii* divides through a process called endodyogeny, where daughter parasites form within the mother parasite. Through this process, some organelles are inherited from the mother parasite, while other organelles are synthesized *de novo* within the daughter parasites [27]. To investigate whether the difference in TgEFP1-HA localization pattern between vacuoles was due to differences in intracellular division stages and to understand the biogenesis of its compartment, we performed IFA of intracellular parasites staining for TgEFP1 and for acetylated-tubulin, which allows us to monitor parasite division. In non-dividing parasites, TgEFP1-HA staining starts as a u-shaped pattern, one per parasite (Figure 2B, top row). During the early division stages, the u-shaped pattern starts to distribute amongst the two daughter cells (Figure 2B, second row). Eventually, the u-shaped pattern turns into two puncta, one per daughter parasite in mid and late division stages (Figure 2B, third and bottom row). This showed that the variation in the TgEFP1-HA localization pattern is attributed to the dynamic pattern of its compartment during division. Furthermore, we showed that the TgEFP1-HA compartment, which appears to be the PLV/ELC, is inherited from the mother parasite.

Interestingly, during inspection of IFAs, we noted a significant amount of TgEFP1-HA signal outside of the parasites, a pattern that was more prominent in mid and late division stages (Figure 2B, white arrow). This suggests that TgEFP1 might also be present within the parasitophorous vacuole (PV). To confirm this observation, we monitored the localization of TgEFP1 by IFA from cultures permeabilized with 0.001% digitonin, which is known to be sufficient to permeabilize the host cell and PV membranes, but not the parasite plasma membrane [19]. Besides anti-HA antibodies to monitor TgEFP1, we stained the partially permeabilized cultures for TgGRA5, which is a known marker for the PV, and for TgNHE3, to confirm that the parasite plasma membrane remained intact. While TgNHE3 is not detectable in partially permeabilized cultures, we can clearly detect both TgGRA5 and TgEFP1 (Figure 2C). Altogether, TgEFP1-HA has a unique dual localization at the PLV/ELC and the PV.

### 3.2. Monitoring of TgEFP1 Localization by Ultrastructure Expansion Microscopy

As to more precisely observe the localization and dynamics of TgEFP1 in intracellular parasites, we employed Expansion Microscopy (U-ExM), which allows for higher resolution of cellular structures and protein localization [20]. Intracellular parasites were treated according to the U-ExM protocol to expand samples [20], and stained for TgEFP1-HA and TgGRA5 using antibodies and for protein density using NHS-ester. Following expansion, the size of the samples increased five-fold, which revealed a high level of detail, not only the localization of TgEFP1-HA, but also the dynamic fragmentation and biogenesis of the compartment within which it localizes (Figure 3). U-ExM images show that TgEFP1-HA localizes to a discrete compartment, as seen in IFAs of non-expanded samples (Figure 3A). Additionally, TgEFP1-HA co-localized with TgGRA5 at the PV in expanded samples, confirming the results described above (Figure 3A). TgGRA5 is also detected within the parasite, which could be the dense granules from which this protein is secreted into the PV. Furthermore, we were able to observe TgEFP1-HA staining in the non-dividing, early, mid, and late division stages by co-staining with acetylated tubulin (Figure 3B). In non-dividing parasites, TgEFP1-HA staining appears as puncta concentrated at a focal area within the parasite. During the early and mid-division stages, we see that TgEFP1-HA starts to distribute amongst the two daughter parasites. Eventually, during late division stages, TgEFP1-HA staining presents as two individual concentrated areas, one per daughter parasite (Figure 3B).

### 3.3. Brefeldin A (BFA) Treatment on Intracellular Parasites Depletes PV Localization but Does Not Affect PLV/ELC Localization of TgEFP1-HA

Trafficking of secretory proteins to endosomal compartments in *T. gondii* is dependent on motor proteins and canonical vesicle transport through the Golgi complex [28]. The regulation of secretory protein trafficking through the Golgi-complex to the endosomal compartments was found to be critical for protein localization to the secretory organelles, as well as for the function of those proteins [28]. To investigate the trafficking of TgEFP1-HA to the PV and its location within the parasite, intracellular parasites were treated with BFA to block ER to Golgi transport, which would inhibit the transport of proteins to endosomal compartments (Figure 4A). IFAs of BFA treated and untreated parasites were stained with anti-HA antibodies to monitor TgEFP1-HA localization, along with acetylated-tubulin to determine the stage of intracellular division (Figure 4B,C). Additionally, the trans-Golgi network protein TgSORTLR was used to confirm the effect of BFA, as treatment inhibits TgSORTLR trafficking to the trans-Golgi network [21]. IFA of intracellular parasites treated with 5 mM BFA for 15 minutes showed that parasites are still expressing TgSORTLR and TgEFP1-HA at intracellular compartments that are abutting each other (Figure 4B, top row). When intracellular parasites were treated with 5 mM of BFA for 1 h, TgSORTLR became undetectable, while the intra-parasitic foci of the TgEFP1-HA signal remained. Interestingly, we observed little to no TgEFP1-HA in the PV of vacuoles in later division stages (Figure 4B, bottom row). In parallel, we treated parasites with 0, 1, 2.5, and 5 mM of BFA for 30 min and performed IFA of treated parasites staining with acetylated-tubulin, TgSORTLR, and TgEFP1-HA (Figure 4C). Parasites treated with 0 and 1 mM of BFA are still expressing TgSORTLR at the trans-Golgi network and TgEFP1-HA within the parasite with some staining at the PV (Figure 4C, top and second row). With increasing BFA concentration to 2.5 and 5 mM, the TgSORTLR signal is diffuse or no longer present. However, TgEFP1-HA expression within the parasite remains with little to no accumulation at the PV (Figure 4C, third and bottom row). Altogether, this showed that transport of TgEFP1-HA to its intra-parasitic localization is not dependent on Golgi-transport.

### 3.4. TgEFP1 Knockout (TgEFP1-KO) Parasites Have a Faster Lytic Cycle

CRISPR/Cas9 was used to knockout TgEFP1 in the TgEFP1-HA expressing parasites to generate TgEFP1-KO parasites [15]. TgEFP1-HA parasites were co-transfected with a plasmid encoding Cas9 and one of two guide RNAs targeting TgEFP1 (sgRNA 1 and 2), along with a donor template consisting of a selectable marker flanked by regions of homology flanking the cut site in the *TgEFP1* locus (Figure 5A). sgRNA 1 targeted the first EF-hand domain, while sgRNA 2 targeted the second EF-hand domain (Figure 5A). In this manner, we established two independent mutant clones, where clone 1 was generated using sgRNA 1, while clone 2 was generated using sgRNA 2. Western blot of protein extracts from these clones showed that they no longer express TgEFP1-HA (Figure 5A). Protein lysate of TgEFP1-HA and parental RHΔKu80 parasites acted as positive and negative controls, respectively. IFA shows that the tagged cell lines TgEFP1 KO clones 1 and 2 are no longer expressing TgEFP1-HA, but are still expressing TgNHE3 in the correct localization (Figure 5B).

While maintaining the parental and mutant strains, we observed that TgEFP1-KO clones 1 and 2 seemed to lyse through a monolayer of HFFs earlier than the parental parasites. To quantitate this effect, we performed plaque assays using the parental strain and the two TgEFP1-KO clones. For all phenotypic assays, we used Rh∆Ku80 as the parental control, as there was no statistical difference between this strain and the TgEFP1-HA line. Representative images of the plaque assays show that TgEFP1-KO clones 1 and 2 have more clearance of the HFF monolayer than the parental parasites (Figure 5C). Quantification of plaque area clearance using Image J software showed that the average percentage plaque area clearance for parental parasites was 14.395% ± 0.875, while that of TgEFP1-KO clones 1 and 2 was significantly higher at 27.79% ± 2.98 and 30.18% ± 6.87, respectively, (Figure 5C).

### 3.5. Parasites Lacking TgEFP1 Are More Sensitive to Ionophore Induced Egress

The increase in cell clearance by the KO strains could indicate a rise in the efficiency or timing of any of the steps of the parasite lytic cycle. Previous studies have shown that calcium fluxes are critical for the regulation of these steps, including the timing of parasite egress during the lytic cycle [6]. As TgEFP1 is predicted to bind calcium via two C-terminal EF-hand domains, we investigated whether the observed faster lytic rate was due to perturbations in calcium-dependent events such as egress. For this purpose, intracellular parasites were treated with 0, 0.1, 0.5, or 1 μM of the calcium ionophore A23187 for 2 min. For each treatment, we calculated the percentage of egressed vacuoles with respect to the no ionophore controls (Figure 5D). At 0.1 mM of A23817, the parental strain did not exhibit any egress (0% ± 0), while TgEFP1-KO clones 1 and 2 showed 23.26% ± 11.26 and 15.37% ± 6.64 egressed vacuoles, respectively. When parasites were treated with 0.5 mM, the trend continued where parental parasites, for which 76.44% ± 4.84 of vacuoles were ruptured, were undergoing less egress than KO clones 1 and 2, which showed 97.21% ± 7.801 and 96.32% ± 7.610 egress, respectively, (Figure 5D). This result shows that parasites lacking TgEFP1 are more sensitive to calcium ionophore treatment in comparison to the parental parasites.

Previously, we have shown that mutants that exhibit increased sensitivity to calcium ionophore-induced egress also show altered dynamics in natural egress, which could be observed by monitoring the sizes of vacuoles along time [14]. For this purpose, we tabulated the percentage of vacuoles of a particular size (i.e., number of parasites per vacuole) at 24 and 30 h post-infection for the parental and KO strains. At 24 h, TgEFP1 KO clones 1 and 2 had a significantly lower percentage of vacuoles with 16 parasites (14.51% ± 3.40 and 9.35% ± 1.83, respectively) in comparison to the parental strain (24.93% ± 4.56) (Figure 5E). Interestingly, at 30 h, TgEFP1-KO clones 1 and 2 had a significantly higher percentage of vacuoles with one parasite (32.71% ± 5.28 and 46.12% ± 9.62, respectively), in comparison to that of both RHΔKu80 and TgEFP1-HA parasites (9.77% ± 3.50 and 18.20 ± 3.96, respectively) (Figure 5E). An increase in the number of vacuoles with just one parasite would indicate that for the KO strains, the parasites are exiting the cell earlier and re-entering cells. Altogether, this showed that TgEFP1-KO clones 1 and 2 have a shorter intracellular division cycle in comparison to TgEFP1-HA parasites.

### 3.6. Mutating the Signal Peptide or EF-Hands Disrupts the Localization Pattern of TgEFP1

To test the role of various domains of TgEFP1 on its localization and function, we used a complementation approach in which we introduced either the wild-type or mutant versions of TgEFP1 into the TgEFP1-KO strains. Unfortunately, the way we created the KO strains used up the best selectable markers available and made complementation challenging. Accordingly, we created a new KO in the RHΔKu80 strain using the same CRISPR/Cas9 strategies and constructs as before. PCR analysis of TgEFP1-KO clone A showed correct insertion of the HXGPRT selectable marker at the endogenous *TgEFP1* locus (Figure 6A). Complemented strains of the TgEFP1-KO Clone A were generated by introducing an ectopic copy of TgEFP1-HA under its endogenous promoter along with a DHFR selectable marker at the Ku80 locus. Complementation was undertaken with either wild-type (TgEFP1 WT), signal peptide truncation (TgEFP1 SP Del), or mutant copies containing point mutations from an aspartate residue at position 97 (TgEFP1 D97A) or 129 (TgEFP1 D129A) to an alanine (Figure 6B). The purpose of complementing with TgEFP1 SP truncation is to understand the function of the signal peptide in the localization of the protein to the PLV/ELC and/or the PV. The purpose of complementing with point mutations at key residues within the EF-hand domains is to understand if the proteins’ predicted ability to bind calcium influences the localization of the protein. IFA analysis showed that TgEFP1-KO clone A parasites complemented with TgEFP1 WT-HA showed a similar localization pattern as that of endogenously tagged proteins at the PLV/ELC and the PV (Figure 6C, left column). In contrast, TgEFP1-KO clone A parasites complemented with TgEFP1 SP Del-HA showed a localization pattern that was cytosolic puncta (Figure 6C, second column). Interestingly, TgEFP1-KO clone A parasites complemented with either TgEFP1 D97A-HA or TgEFP1 D129A-HA showed a localization pattern that was exclusively in the PV (Figure 6C, third and right column). Acetylated tubulin staining showed that all representative vacuoles are non-dividing and outlined the cell body of the parasite to allow for differentiation of localization patterns that are intra-parasitic versus extra-parasitic (Figure 6C, mid and bottom row).

Importantly, the new knockout clone TgEFP1 KO clone A showed the same plaquing phenotype as KO clones 1 and 2 (Figure 6D). TgEFP1 KO Clone A had a significantly higher average percentage of plaque area clearance in comparison to that of the parental strain (52.10% ± 18.73 vs. 37.13% ± 10.24). This phenotype was complemented by the introduction of a wild-type copy of TgEFP1 (29.57% ± 5.00, Figure 6D). Indeed, there was no significant difference between the plaquing efficiency of the parental strain and complemented strain (Figure 6D). Interestingly, the three strains complemented with mutant versions of TgEFP1 propagate at a significantly lower rate than the knockout strain, which would suggest complementation of the mutant phenotype (Figure 6D). Nonetheless, these three mutant-complemented strains are significantly less efficient at forming plaques than both the parental strain and the WT complemented strain. Thus, it is plausible that by being mis-localized and/or non-functional, these mutant versions of TgEFP1 are imparting a dominant-negative effect.

## 4. Discussion

Calcium, one of the most pervasive second messengers, regulates vital cellular functions, including gene expression, protein secretion, metabolism, and apoptosis in a wide variety of cells [1]. For this reason, cells utilize several calcium homeostatic mechanisms to regulate calcium fluxes under resting and non-resting conditions [1]. These mechanisms include activation of calcium-sensing and/or conducting proteins, which leads to release or uptake of calcium by major calcium reservoirs within the cell or from the extracellular environment [1]. Furthermore, calcium-binding proteins can act as buffers to regulate the duration of calcium fluxes.

Out of all the calcium-binding proteins, EF-hand domain-containing ones are the most abundant [29]. The structural composition of EF-hand domains consists of a Ca^2+^-coordinating loop that is flanked by two alpha-helices [1,30]. Specifically, the Ca^2+^-coordinating loop comprises 5–7 ligands, which are primarily carboxylate groups arranged in a pentagonal bipyramid [30]. Most frequently, two EF-hand domains are paired to form the functional calcium-binding unit [31]. EF-hand domains are categorized as either canonical or pseudo domains [31]. Canonical EF-hand domains consist of pairs of EF-hand motifs that work in concert to coordinate calcium-binding through carboxylate groups of the residues that comprise them. In contrast, pseudo-EF-hand domains consist of a single and/or an odd number of EF-hand motifs that bind calcium through backbone carbonyl groups [30]. Due to the difference in structures between these two classes of EF-hand domains, there is a difference in calcium-binding affinities [1,31]. Specifically, canonical EF-hand domains have a relatively higher calcium-binding affinity than that of pseudo-EF-hand domains [1,31]. This leads to two main functional classes of EF-hand domain-containing proteins: calcium-sensing, which are mostly comprised of pseudo-EF-hand domains, and calcium buffering proteins, which are mostly comprised of canonical EF-hand domains [1,31]. Furthermore, calcium buffering EF-hand domain-containing proteins exhibit limited conformational changes upon calcium binding, whereas calcium-sensing EF-hand domain-containing proteins exhibit large conformational changes upon calcium binding to allow for interacting proteins to bind [31]. For these reasons, calcium buffering EF-hand domain-containing proteins primarily play a role in modulating the duration of calcium signaling and maintaining calcium homeostasis, while calcium-sensing EF-hand domain proteins primarily respond to physiological changes in calcium [1].

Just as in mammalian and plant cells, calcium also plays a key role in regulating the growth cycle of the eukaryotic human pathogen *Toxoplasma gondii* [3]. Not surprisingly, *Toxoplasma* encodes numerous proteins containing EF-hand domains. Previous studies have identified 68 EF-hand domain-containing proteins within the *Toxoplasma* genome, of which eight contain transmembrane domains [32]. Among the most studied EF-hand proteins in *Toxoplasma* are the calcium-dependent protein kinases (CDPKs). CDPKs, which are unique to plants and some protozoan parasites and absent in mammalian cells, contain a serine/threonine kinase domain and a calmodulin-like domain linked by an autoinhibitory junction domain [33]. In *Toxoplasma*, CDPKs have been implicated in a diversity of functions, including secretion, motility, invasion, egress, and parasite division [34,35,36]. Other categories of EF-hand domain-containing proteins characterized in *Toxoplasma* fall into the functional categories of calmodulin (CaM), CaM-like, centrins, calcatrins, and calcineurin proteins [3]. Given the great diversity of functions imparted by EF-hand proteins and the importance of calcium signaling in this parasite, there is a need to characterize a broader range of EF-hand domain-containing proteins.

We have now identified and characterized TgEFP1, which, based on *in silico* analysis of its primary and secondary structure, has a predicted N-terminal signal peptide and two EF-hand domains located at the C-terminus of the protein (Figure 1A). In-depth analysis of the Ca^2+^ coordinating residues shows that the EF-hand domains contain 12 of them, including two aspartic acid residues at positions 97 and 129, suggesting that TgEFP1 contains canonical EF-hand domains. For this reason, it is predicted that TgEFP1 acts as a calcium-buffering protein and may play a role in modulating the duration of calcium fluxes and maintaining calcium homeostasis in the parasite. Furthermore, no transmembrane domains and/or post-translational modification were predicted that would suggest the protein was membrane-associated. Therefore, TgEFP1 would be the first identified luminal protein of the PLV/ELC.

Interestingly, TgEFP1 can be detected both in the PLV/ELC and within the parasitophorous vacuole (PV). To our knowledge, this is the first report of a *Toxoplasma* protein that localizes to both the PLV/ELC and the PV, suggesting a crosstalk between these two organelles. Previous studies have identified crosstalk between the endocytic and exocytic system, where the VAC played a key role in the export and import of proteins from the PV and host cell to the parasite cell body [37]. Since the VAC and other compartments of the endo-lysosomal system, including the PLV/ELC, can fuse in extracellular parasites, it is possible that there is an exchange of contents between all of these compartments. Thus, the dual localization of TgEFP1 may be attributed to interactions between *Toxoplasma’s* endocytic and exocytic systems.

Lack of TgEFP1 results in disruption of the normal propagation of the parasite, as well as an altered sensitivity to calcium ionophores. These phenotypes would suggest that TgEFP1 plays a role in calcium-dependent processes and/or homeostasis. As TgEFP1 is in both the PV and the PLV/ELC, a question arises as to in which of those two locations is TgEFP1′s function being imparted. Interestingly, the phenotypic defects observed in the TgEFP1 knockout parasites parallel those of parasites lacking the PV localized dense granule protein GRA41 [14]. GRA41 is a dense granule protein that is secreted into the PV, and loss of GRA41 leads to defects in calcium regulation and timing of egress [14]. The fact that loss of either GRA41 or TgEFP1 results in premature egress might suggest that events at the PV might act as a hub for regulation of parasite exit and a plausible locale for TgEFP1′s function.

Nonetheless, it is also plausible that TgEFP1 acts within the PLV/ELC. Lysosomes, endosomal vesicles, peroxisomes, and secretory vesicles can function in calcium release to induce signaling that regulates localized cellular functions, including protein transport, protein secretion, and vesicle fusion [38]. Unique to plant cells are EF-hand domain containing proteins that localize to the plant vacuole [39,40]. The EF-hand domain containing proteins that are specific to plant vacuoles play a key role in mediating calcium signaling that is implicated in osmoregulation under stress conditions, including drought, cold, and salt stress [39]. Furthermore, plant EF-hand domain-containing proteins regulate calcium signaling, which is vital to plant adaptive behavior [39]. Lastly, the plant vacuole has been shown to be a major calcium reservoir within plant cells, where the entry and release of calcium are regulated by the EF-hand domain-containing proteins that localize to the plant vacuole [39,40]. Therefore, the PLV/VAC may also play a role as a calcium reservoir in the parasite, and TgEFP1 may play an important role in adaptive behaviors and osmoregulation in the parasite.

To start addressing the relation between function and localization for TgEFP1, we investigated the consequence of mutating the signal peptide. Unfortunately, this mutant protein was not localized to either the PLV/ELC or the PV and appeared to cause a dominant negative effect. Interestingly, when we mutated either of the EF-hand domains, TgEFP1 was only present in the PV. This suggests that the binding of calcium by both EF-hand domains is necessary for the localization of the protein to the PLV/ELC. A plausible model that could explain this result is that that TgEFP1 is initially secreted to the PV, and upon binding to calcium, it is internalized to the PLV/ELC as a mechanism to regulate calcium homeostasis within the parasite. This model is consistent with our observations from the experiments using BFA, which showed that disruption of Golgi-dependent trafficking affected TgEFP1-HA localization to the PV. However, the apparent lack of TgEFP1-HA in the PV upon BFA treatment could be due to an effect in PLV/ELC vesical transport and fusion. Lastly, the endocytic and exocytic systems of the parasite are known to converge at endosomal compartments to regulate the import and export of proteins from the parasite cell body [37]. Thus, through the function of TgEFP1, the PV and the endosomal compartments might be collaborating in regulating calcium homeostasis and facilitating calcium signaling.

In terms of function, TgEFP1 may act as a calcium buffer that actively regulates levels of calcium within the PV, parasite cytosol, and the PLV/ELC. Specifically, TgEFP1 may initially be transported to the PV, where it binds to excess calcium. Upon binding to calcium, the protein is internalized through the function of the PLV/ELC. The calcium is then retained in the PLV/ELC to also regulate calcium levels within the parasite cytosol. Future directions will be to identify and characterize TgEFP1 interactors that also localize to the PLV/ELC to gain a better understanding of its function in the parasite. Lastly, calcium studies, including the use of live calcium indicators, could determine the functional relevance of TgEFP1 indirectly regulating calcium fluxes within the parasite cytosol and the PV during the lytic cycle of the parasite.

## Figures and Tables

**Figure 1 cells-11-01709-f001:**
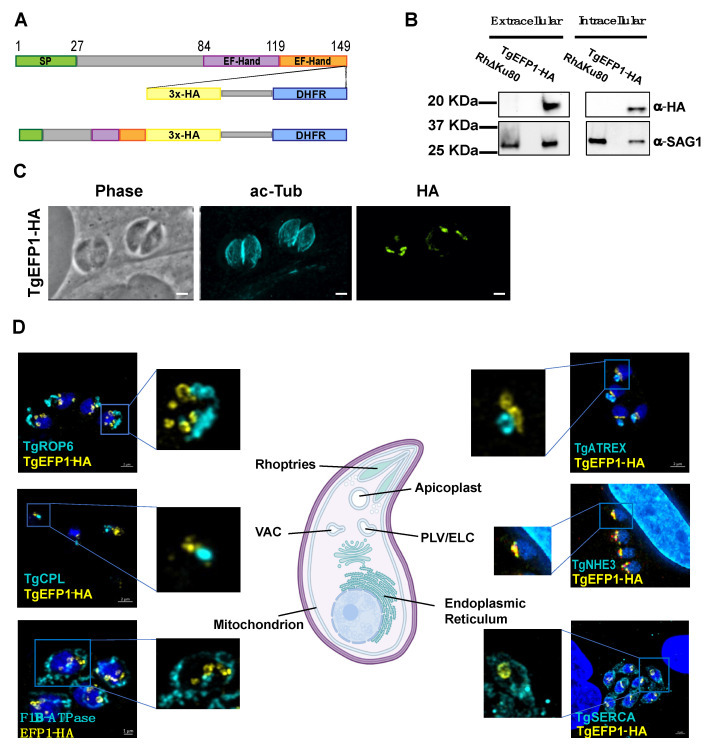
TgEFP1 expression and localization in intracellular and extracellular parasites. (**A**) Diagram of TgGT1_255660 (TgEFP1), which contains an N-terminal signal peptide (green box) along with two consecutive C-terminal EF-hand domains (purple and orange boxes). Double homologous recombination was used to introduce a triple hemagglutinin tag (3xHA, yellow box) and a DHFR selectable marker (DHFR, blue box) to the C-terminus of TgEFP1. (**B**) Western blot analysis of protein extract from both intracellular and extracellular parasites of the parental and the TgEFP1-HA strains. Blots were probed for HA (top panel) and for TgSAG1 as a loading control (bottom panel). (**C**) IFA of intracellular parasites of the TgEFP1-HA strain probing with anti-HA (green) and anti-acetylated-tubulin (cyan). (**D**) IFA of intracellular TgEFP1-HA parasites using anti-HA antibodies (green) and antibodies against known organellar markers (cyan), including ROP 6 for the rhoptries, TgCPL for the VAC, F1B-ATPase for the mitochondrion, TgATREX for the apicoplast, TgNHE3 for the PLV, and TgSERCA for the ER (Image created with BioRender.com). Scale bars = 2 mm.

**Figure 2 cells-11-01709-f002:**
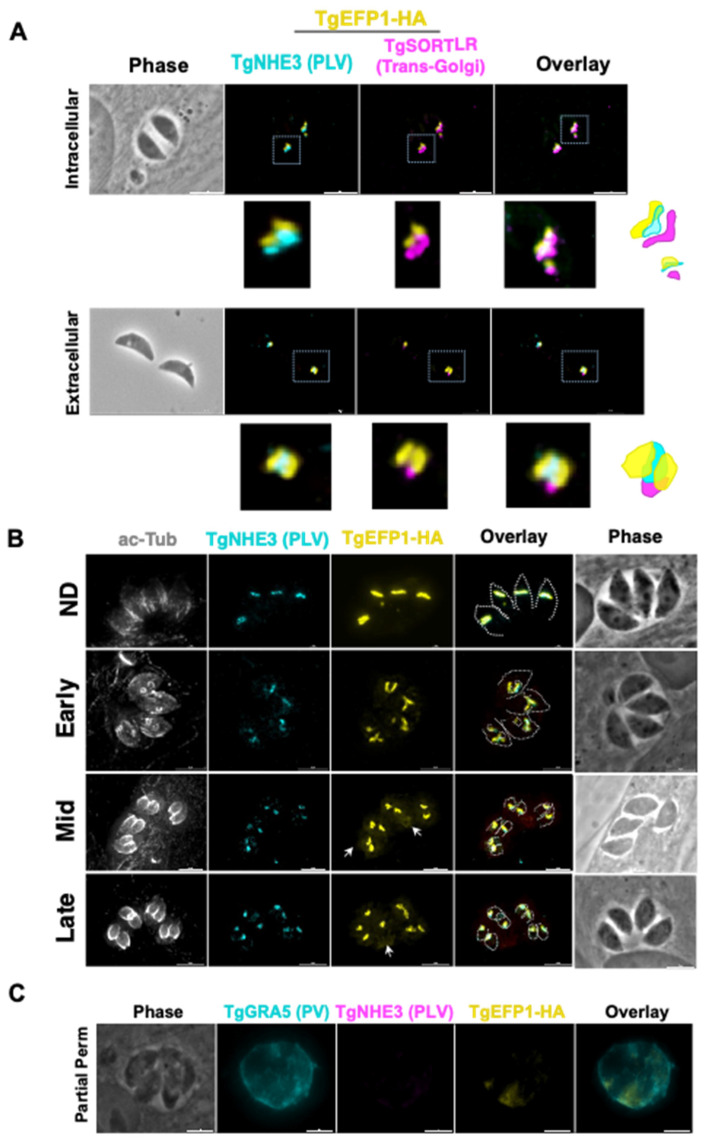
Dynamics of TgEFP1-HA localization during division. (**A**) IFA of intracellular and extracellular TgEFP1-HA parasites probed with HA antibodies (yellow), the PLV/ELC marker TgNHE3 (cyan), and the trans-Golgi marker-TgSORTLR (magenta). The white box marks area zoomed on in the images below. The diagram on the right shows the relative position of the three staining patterns. (**B**) IFA of intracellular parasites using antibodies against acetylated tubulin (ac-Tub), TgNEH3, and the HA epitope tag (yellow). Acetylated tubulin was used to categorize the stage of division as either non-dividing (ND), early, mid, or late. The white dashed tracing delineates mother and daughter parasite ac-tubulin structure. The white arrows indicate TgEFP1-HA localization in the PV. (**C**) IFA of intracellular TgEFP1-HA parasites using partial permeabilization stained for the parasitophorous vacuole (PV) marker TgGRA5 (cyan), HA (yellow), and TgNHE3. The lack of TgNHE3 staining confirms that only the host and parasitophorous vacuole membranes were permeabilized but not that of the parasite. Scale bar = 5 mm.

**Figure 3 cells-11-01709-f003:**
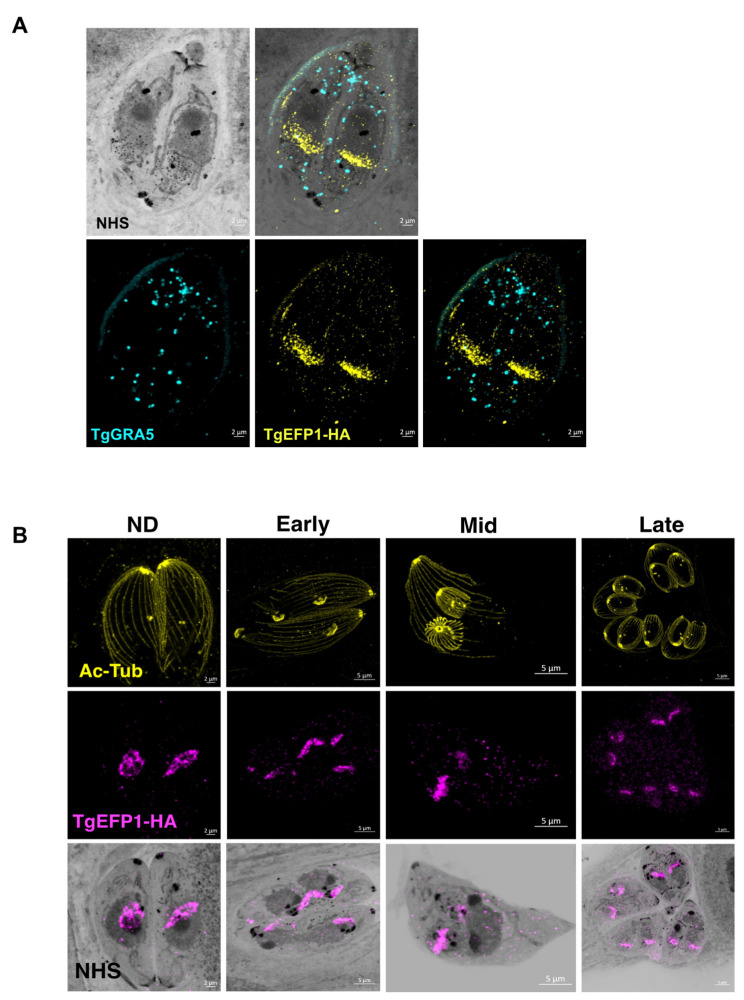
Ultra-structure expansion (U-ExM) microscopy of TgEFP1-HA intracellular parasites. (**A**) Intracellular TgEFP1-HA parasites were processed for U-ExM and stained with NHS-ester (inverted grayscale), TgGRA5 (cyan), and TgEFP1-HA (yellow). (**B**) Images show U-ExM of intracellular TgEFP1-HA parasites stained with NHS-ester (inverted grayscale), acetylated-tubulin (ac-Tub, yellow), and TgEFP1-HA (magenta). The stages of parasite division were categorized as non-dividing (ND), early, mid, and late, using Ac-Tub as a guide. The bottom row shows the overlay of NHS-ester and TgEFP1-HA. Scale = 2 µm.

**Figure 4 cells-11-01709-f004:**
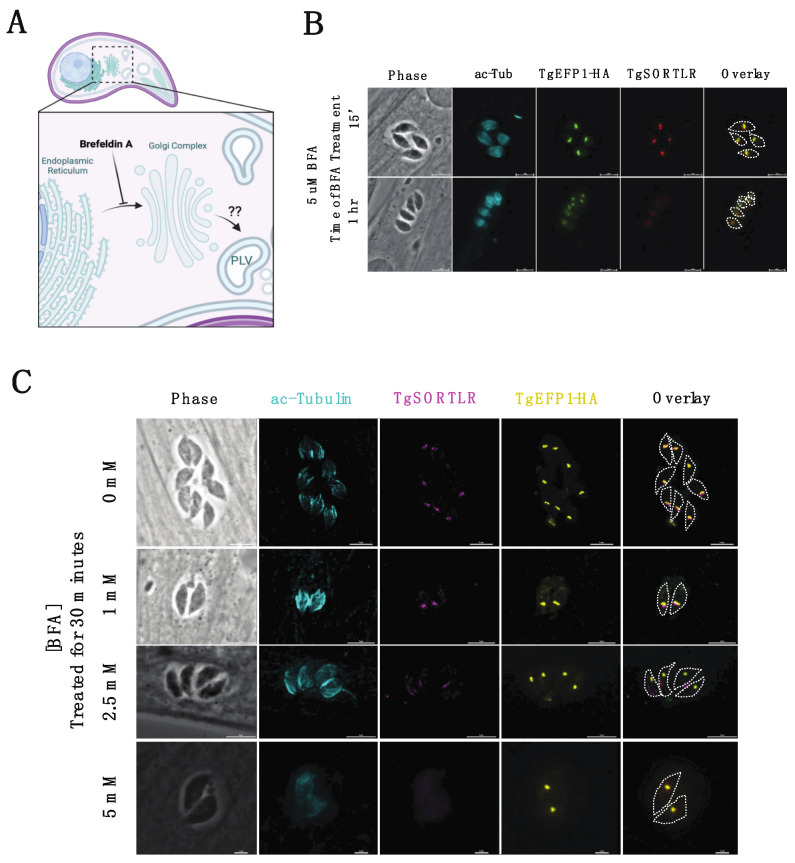
Effect of Brefeldin A treatment on TgEFP1-HA localization in intracellular parasites. (**A**) Diagram showing the effect of Brefeldin A (BFA) on protein transport and localization (created with BioRender.com). BFA inhibits vesicular fusion and transport of proteins from ER to the Golgi complex. (**B**) IFA of intracellular parasites treated with 5 mM BFA for 15 min (top row) or 1 h (bottom row) stained for TgSORTLR (red), TgEFP1-HA (green), acetylated-tubulin (cyan). (**C**) IFA of intracellular parasites treated with a gradient concentration of BFA from 0 mM to 5 mM for 30 min and stained as in A. The white dashed line delineates parasite cell body. Scale bar = 5 μm.

**Figure 5 cells-11-01709-f005:**
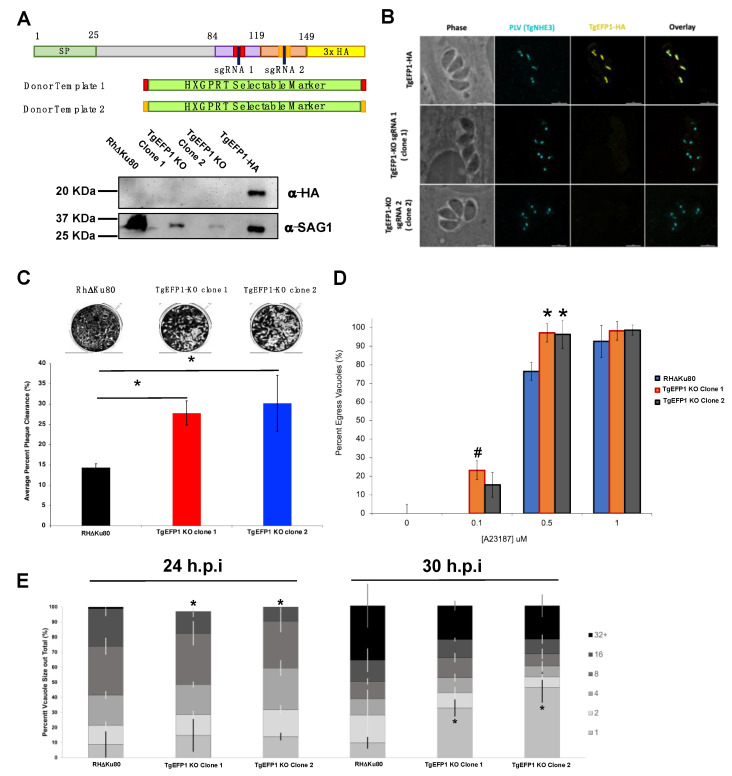
Generation and phenotypic analysis of TgEFP1 knockout (TgEFP1-KO) parasites. (**A**) Diagram showing the CRISPR/Cas9 strategy used to knockout (ko) TgEFP1 in the TgEFP1-HA strain. Two guide-RNAs (sgRNA 1 and sgRNA 2) were designed to target each of the EF-hand domains of TgEFP1. Two donor templates (1 and 2) were designed to contain the HXGPRT selectable marker flanked by regions of homology to the targeted area (red and orange boxes). The vector coding Cas9 and sgRNA1 and the donor template 1 were co-transfected to generate TgEFP1 Clone 1 and sgRNA2, and the donor template 2 were co-transfected to generate TgEFP1 Clone 2. Below the diagram is a western blot of protein extract from the parental RHDKu80 strain, the two independent TgEFP1-KO clones, and the TgEFP1-HA parasites probed for either HA or SAG1 (loading control). (**B**) IFA of intracellular parasites of the TgEFP1-HA, TgEFP1-KO clone 1, and TgEFP1-KO clone 2 strains probed for HA (yellow) and TgNHE3 (cyan). Scale bar = 5 μm. (**C**) Representative images of plaque assays of RH∆Ku80 and KO clones 1 and 2 grown for 5 days post-infection. The graph shows the quantification of plaque area clearance. (**D**) Intracellular parasites of the three strains were treated with 0, 0.1, 0.5, or 1 μM of calcium ionophore-A23187 for 2 min, and the percent of lysed vacuoles was calculated for each data point. (**E**) Doubling assays show that at 24 h, there was a significantly lower number of 16 packs in comparison to parental and tagged parasites. At 30 h, TgEFP1-KO clones 1 and 2 had a significantly higher number of 1 pack in comparison to that of parental and tagged parasites. For C to E n is 3 experimental replicates with experimental triplicates each, error bars are standard deviation; * *p* < 0.05 and # *p* < 0.01 based on Student’s *t*-test.

**Figure 6 cells-11-01709-f006:**
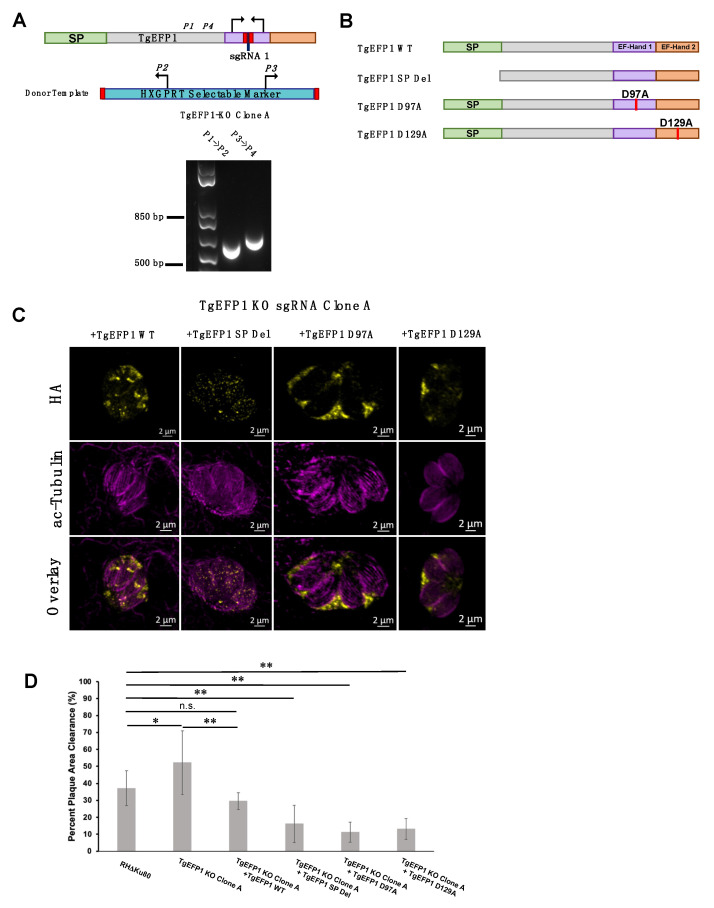
Role of TgEFP1 domains in localization and function. (**A**) Diagram of CRISPR/Cas9 strategy to generate TgEFP1-KO parasites in RH∆Ku80. The sgRNA used was designed to target the first EF-hand domain (purple box). P1-4 indicates the position and direction of the primers used to confirm the integration of the donor template. Shown below the diagram is the result of the PCR analysis of genomic DNA from TgEFP1-KO clone A using primer ss1 and 2 or primer 3 and 4. (**B**) Diagrams of versions of TgEFP1-HA used to complement the knockout strain. Wild-type TgEFP1-HA is shown on top. Amino acids 1 to 28 were deleted to generate the signal peptide deletion (SP Del) mutant. To disrupt the EF-hands, either the aspartic acid (D) at position 97 within EF-hand 1 or at position 129 within EF-hand 2 were mutated to alanine (A) to generate the D97A and D129A mutants. (**C**) IFA of intracellular parasites of the TgEFP1 KO Clone A strain complemented with either wild-type (+TgEFP1 WT), signal peptide deletion (+TgEFP1 SP Del), D97A (+TgEFP1 D97A), or D129A (+TgEFP1 D129A). TgEFP1 KO clone A parasites complemented TgEFP1 WT shows TgEFP1-HA (yellow) localization at the PLV and PV. TgEFP1 KO clone A parasites complemented with TgEFP1 SP Del shows TgEFP1-HA (HA; yellow) localized at intracellular puncta. TgEFP1 KO clone A parasites complemented with TgEFP1 D97A or D129A localized exclusively to the PV. All samples were co-stained with acetylated-tubulin (ac-tubulin; magenta) to ensure all representative images were of non-dividing parasites and to differentiate intra- and extracellular compartments. (**D**) Plaque assay of the parental RHDKu80, TgEFP1 KO clone A, and all complement cell lines. Scale bar = 2 μm. * *p* < 0.05 and ** *p* < 0.01, n.s. = no significant difference based on ANOVA followed by post-hoc Student *t*-tests.

## Data Availability

Not applicable.

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
