# Peer review of "The Dually Localized EF-Hand Domain-Containing Protein TgEFP1 Regulates the Lytic Cycle of *Toxoplasma gondii"

_cells, 2022, doi:10.3390/cells11101709_

Round 1
Reviewer 1 Report
This as important contribution to a better knowledge of the intracellular life cycle of Toxoplama gondii. The authors characterized a novel EF-hand domain-containing protein that localizes in structures such as the plant-like vacuole, the endo-lysosomal compatment, and also within the parasitophorous vacuole. This protein is in some way associated with the dynamics of parasite proliferation and egress from the host cell. Surprisingly no comments are made to the acidocalcisome, a organelle showed to be involved in calcium regulation in this pathogenic protozoan.
Author Response
We have now added information about the acidocalcisomes within the introduction of the manuscript draft, along with references for the role of acidocalcisome protein function in the parasite.
Reviewer 2 Report
In this manuscript, Noopur Dave describes the identification of a novel Toxoplasma EF-hand domain-containing protein, named TgEFP1, that binds calcium. The calcium signaling pathway is involved in various biological processes in T. gondii, including tachyzoite egress from the host. The subcellular localization of TgEFP1 is shown. Based on a KO assay, it is shown that TgEFP1 would have a role in modulating the egress of Toxoplasma from the host. It is also shown that mutations in the EF-Hand domains lead to mislocalization of the protein and impart a dominant-negative effect.
The work is well done, the subcellular localization was thoroughly analyzed and the biological role experiments allow the authors to establish a relevant and interesting function of TgEFP1 as a calcium buffering protein, modulating the duration of calcium flux.
Minor comments:
- Figure 1B. An overlay of TgEFP1 and acTub would be useful to more clearly determine their location during cell division.
- Fig 4 and line 331, The authors mention that no TgEFP1 marking is observed in PV in later stages of division. It is really difficult to perceive the mark on the PV in all cases. Beyond an improvement in the image, a delimitation of the intracellular tachyzoites would help to see the mark in the lumen of the PV.
- The loss of the TgEFP1 gene induces a rapid egress of the tachyzoite from the host, in such a way that PV with a high number of tachyzoites are hardly observed. Could differences in intensity levels (expression) or a mislocalization of TgEFP1 be observed in PVs with 32 or more tachyzoites?
Author Response
Response to minor concerns are as follow.
- We are assuming that the reviewer is referring to 2B . We have now delineated the cell bodies of mother and daughter parasites in the overlay images as to highlight the position of the staining relative to the body of the parasites.
- We have delineated the cell bodies of the parasites in this figure as we did for figure 2.
- We do not see any differences in the intensity of TgEFP1 staining within the PLV/ELC as the vacuole become bigger. By contrast, TgEFP1 localization to the PV is proportional to the number of parasites in the vacuole (i.e. the more parasites the more staining). As this is typical of most proteins constitutively secreted into the PV it is difficult to correlate that phenomenom to function.